# Identifying Critical Binder Attributes to Facilitate Binder Selection for Efficient Formulation Development in a Continuous Twin Screw Wet Granulation Process

**DOI:** 10.3390/pharmaceutics13020210

**Published:** 2021-02-03

**Authors:** Lise Vandevivere, Maxine Vangampelaere, Christoph Portier, Cedrine de Backere, Olaf Häusler, Thomas De Beer, Chris Vervaet, Valérie Vanhoorne

**Affiliations:** 1Laboratory of Pharmaceutical Technology, Ghent University, Ottergemsesteenweg 460, 9000 Ghent, Belgium; Lise.Vandevivere@UGent.be (L.V.); Maxine.Vangampelaere@gmail.com (M.V.); Christoph.Portier@UGent.be (C.P.); Cedrine.deBackere@UGent.be (C.d.B.); Chris.Vervaet@UGent.be (C.V.); 2Roquette Frères, Rue de la Haute Loge, 62136 Lestrem, France; Olaf.Haeusler@roquette.com; 3Laboratory of Pharmaceutical Process Analytical Technology, Ghent University, Ottergemsesteenweg 460, 9000 Ghent, Belgium; Thomas.DeBeer@UGent.be

**Keywords:** continuous manufacturing, wet granulation, twin-screw granulation, pharmaceutical binders, granule quality, tablet quality, formulation

## Abstract

The suitability of pharmaceutical binders for continuous twin-screw wet granulation was investigated as the pharmaceutical industry is undergoing a switch from batch to continuous manufacturing. Binder selection for twin-screw wet granulation should rely on a scientific approach to enable efficient formulation development. Therefore, the current study identified binder attributes affecting the binder effectiveness in a wet granulation process of a highly soluble model excipient (mannitol). For this formulation, higher binder effectiveness was linked to fast activation of the binder properties (i.e., fast binder dissolution kinetics combined with low viscosity attributes and good wetting properties by the binder). As the impact of binder attributes on the granulation process of a poorly soluble formulation (dicalcium phosphate) was previously investigated, this enabled a comprehensive comparison between both formulations in current research focusing on binder selection. This comparison revealed that binder attributes that are important to guide binder selection differ in function of the solubility of the formulation. The identification of critical binder attributes in the current study enables rational and efficient binder selection for twin-screw granulation of well soluble and poorly soluble formulations. Binder addition proved especially valuable for a poorly soluble formulation.

## 1. Introduction

The pharmaceutical industry is undergoing a gradual switch from batch to continuous manufacturing caused by multiple drivers, including reduction of cost, improved process efficiency, better control strategies, and lowering the environmental footprint [1,2,3,4,5]. Transition to continuous manufacturing is often based on the same formulation as used for batch processing in order to avoid additional studies [6]. However, as batch processing is fundamentally different from continuous processing (e.g., material residence time), materials are not necessarily suitable for both processing techniques [7]. Therefore, the suitability of different materials for continuous manufacturing needs to be investigated.

Twin-screw granulation (TSG) has proven its efficacy in continuous solid-dosage manufacturing, as this technique allows continuous production of granules while offering flexibility towards granule properties [8,9,10,11,12,13]. Continuous TSG also requires minimum effort for scale-up, as batch sizes are determined by adjusting the total process time, saving time and costs. Next to the process parameters (e.g., screw speed, material throughput, barrel temperature), product quality is strongly influenced by the formulation variables in TSG [11,14,15]. Binders are often included in the formulation to facilitate the granulation process (e.g., granule nucleation and growth). As the binder type affected granule quality [10,16,17], binder selection for continuous TSG should rely on a scientific approach to obtain efficient formulation development. This starts with a profound understanding of the raw material attributes influencing the granulation process because knowledge of attributes helps to predict their impact on functionality. According to the quality-by-design approach, one has to identify critical binder attributes, as these impact tablet and granule quality attributes (e.g., in terms of tablet tensile strength, granule friability) [18,19]. Some studies have already reported on the influence of binder attributes upon granule quality for TSG. Principal component analysis (PCA) was used by Willecke et al. to identify the attributes that were responsible for (dis)similarities between binders. However, the binder characterization mainly included only properties of binder dispersions, and the variability between binders was described by the binder’s viscosity properties, surface tension and glass transition temperature. It was concluded that these binder attributes impacted the granule quality [14,20]. In addition, Dhenge et al. investigated the effect of the properties of the granulation liquid on granule attributes. High binder viscosity positively affected the granule quality, whereas the surface tension of the granulation liquid had only a minor influence. However, only a single type of binder (hydroxypropyl cellulose (HPC)) was used to confirm these observations [21]. Other studies have assessed the binder types’ effect (e.g., hydroxypropyl methylcellulose (HPMC), polyvinylpyrrolidone (PVP), HPC, native and pregelatinized starches, maltodextrins, and vinylpyrrolidone-vinyl acetate copolymer) on granule quality for TSG [9,10,17,20,22,23,24,25,26,27,28,29]. Nevertheless, the use of different formulations is hindering the comparison on binder efficiency between different studies, as the other components also contributed to the binding performance, affecting the granulation process [30].

In order to select appropriate binders for formulation design in continuous TSG, a systematic and extensive binder characterization is needed. In previous research [16], different binder types were characterized, and their attributes were linked to binder effectiveness using a poorly soluble model excipient (dicalcium phosphate). Binder effectiveness was defined as the amount of liquid required for binder activation in order to achieve good granule friability (<30%) [16]. As the formulation solubility defines the granulation mechanism [11,29], the binder attributes affecting the granule quality of a highly soluble formulation (mannitol) were examined in the current paper. This formulation was used as a model formulation, limiting the interaction with other possible components in the granulation process, affecting binder performance. This way, the binder’s effect could be profoundly studied. The produced granules were also compacted, and the quality of the resulting tablets was assessed. Additionally, appropriate binder selection was compared for a poorly (DCP) and highly soluble (mannitol) formulation. This way, the current study promotes rational binder selection for a continuous wet granulation process. Dry binder addition was applied in this research, as it has already proven its efficiency [16]. Granule quality was determined by measuring the granule friability as an indicator of granule strength. This term should, however, not be confused with pharmaceutical friability, as with the latter, the friability of a tablet formulation is intended.

## 2. Materials and Methods

### 2.1. Materials

Mannitol 50 C (Pearlitol^®^ 50 C, Roquette Frères, Lestrem, France) was used as a highly soluble model excipient. Hydroxypropyl (HP) pea starch (Lycoat^®^ RS 720, Roquette Frères, Lestrem, France), HPMC E5 and E15 (Methocel^®^ E5 and E15, Dow Chemical Company, Rheinmünster, Germany), maltodextrin with a DE of 12, obtained from maize starch (Lycatab^®^ DSH, Roquette Frères, Lestrem, France), maltodextrin with a dextrose equivalent (DE) of 2 and 6, obtained from waxy maize starch (Glucidex^®^ 2 and 6, Roquette Frères, Lestrem, France), polyvinyl alcohol (PVA) 4-88 (Parteck^®^ MXP, Merck Darmstadt, Germany), PVP K12, K30 and K90 (Kollidon^®^ K12, K30 and K90, BASF, Ludwigshafen, Germany), and sodium octenyl succinate starch (SOS) CO 01 (Cleargum^®^ CO 01, Roquette Frères, Lestrem, France) were used as pharmaceutical binders. Magnesium stearate (MgSt VEG E 470b, WIGA Pharma, Hamburg, Germany) was used as a lubricant during tableting.

### 2.2. Methods

#### 2.2.1. Preparation of Granules

First, different preblends were prepared of the highly soluble excipient (mannitol) (95% *w*/*w*) with different binders (HPMC E15, HP pea starch, maltodextrin 6, PVP K12, PVP K90, PVA 4-88, and SOS CO 01) (5% *w*/*w*) using a 20 L tumbling blender (Inversina Bioengineering, Wald, Switzerland), for 15 min at 25 rpm. Drum filling level was each time approximately 75%. These preblends were gravimetrically fed (KT20, K-Tron Soder, Niederlenz, Switzerland) into the co-rotating twin-screw granulator (length-to-diameter (L/D) ratio of 20/1), which is part of a continuous from-powder-to-tablet line (ConsiGma^TM^-25 system, GEA Pharma Systems, Wommelgem, Belgium) at a throughput of 20 kg/h. Before performing experiments, demineralized water was equilibrated at room temperature and used as granulation liquid. The granulation liquid was added to the granulator barrel immediately before the first kneading zone by using two silicon tubes, both connected to a 1.6 mm nozzle and two out-of-phase peristaltic pumps (Watson Marlow, Cornwall, UK). Liquid-to-solid (L/S) ratios ranging from 0.055 to 0.105 (at five different levels) were used. The screw configuration consisted of two kneading zones, separated by a 1.5 L/D-ratio conveying element. The kneading zones were each made up of six kneading elements (0.25 L/D-ratio) in a forward stagger angle of 60°. After the second kneading block, conveying and size control elements were added, both having a length of 3.0 L/D-ratio [17]. The screw speed was set at 500 rpm. The barrel jacket was equipped with an active cooling system in order to maintain the temperature during processing at 30 °C. The torque was monitored by a built-in torque-gauge at 1-s intervals. The torque was not a limiting factor for any of the granulation experiments with the binders. Wet granules were sampled (800 g) at the outlet of the granulator after torque stabilization (i.e., a maximal deviation of 1.0 Nm during one minute prior to sampling), and consequently, tray dried at 40 °C. A moisture analyzer including a balance and an infrared dryer (Mettler LP16, Mettler-Toledo, Zaventem, Belgium) was used for determining the loss-on-drying (LOD) value by drying 1 g of granules at 105 °C until the weight was constant for 30 s. Granules were defined as being dry when an LOD value between 1% and 3% was reached.

Granulation experiments with DCP as poorly soluble model excipient were previously performed by Vandevivere et al. [16]. These experiments were conducted at similar process conditions as for the mannitol formulation. However, in order to obtain efficient granulation with DCP, a higher barrel fill was required to obtain sufficient material densification because of the poor aqueous solubility and highdensity of the excipient. This was obtained by a low screw speed (300 rpm) and a high material throughput (20 kg/h). Furthermore, for DCP, the L/S-ratios depended on the binder (Table 1) using 4 levels, except for PVA 4-88 (2 levels), SOS CO 01 (3 levels) and maltodextrin 6 (3 levels) due to over-wetting of the material at higher L/S-ratio. A comparison between the mannitol and the DCP formulation was addressed in the current study.

#### 2.2.2. Granule Evaluation

Prior to friability testing, all samples (800 g) were split using a rotary cone sample divider (Laborette 27, Fritsch Idar-Oberstein, Germany) to ensure representative sampling. Granule friability was subsequently measured in triplicate for the produced granules, as this measurement illustrates the granule strength. Prior to analysis, the granule fraction >250 µm was separated by sieving. 10 g (*m*_1_) of this fraction was added to a plexiglass drum with baffles together with 200 glass beads (mean diameter of 4 mm) (Carl Roth, Karlsruhe, Germany) and attached to a friabilitor (PTFE Pharma Test, Hainburg, Germany). The drum was rotated at a speed of 25 rpm for 10 min. Afterward, the granules and glass beads were separated. The granular mass >250 µm was again determined (*m*_2_). The granule friability (%) was calculated according to Equation (1):(1)$$Granule\;friability\;\left(\%\right)=\frac{\left(m_{1}-m_{2}\right)}{m_{1}}\times 100$$

#### 2.2.3. Preparation of Tablets

Prior to tableting, granules produced at an L/S-ratio of 0.08 were milled through a 1500 µm grater screen with a square impeller at 900 rpm using the Quadro comill (U10, Quadro, Ontario, Canada) incorporated in the ConsiGma™-25 line. The milled granule fraction of 150–850 µm was blended with 0.7% magnesium stearate in a tumbling blender for 5 min at 49 rpm (Turbula T2F, WAB, Muttenz, Switzerland) before tableting. Tablets were produced on a rotary tablet press simulator (STYLCAM200 R, Medelpharm, Beynost, France) equipped with one pair of flat-faced Euro B punches of 10 mm diameter (Elizabeth Europe, La Chaussee Saint-Victor, France). Compression tests were carried out at a compression speed of 10 tablets/min and at main compaction pressures of 64, 127 and 191 MPa with a targeted tablet weight of 400 mg. At each compaction pressure, 50 tablets were produced.

#### 2.2.4. Tablet Evaluation

The hardness, thickness and diameter of tablets (n = 10) were determined using a hardness tester (ST50, Sotax, Saint-Louis, France). The tensile strength of tablets was calculated according to the formula of Fell and Newton (2) [31]:(2)$$Tablet\;tensile\;strength=\frac{2F}{\pi dt}$$
where *F*, *d* and *t* denote the diametral crushing force, tablet diameter and tablet thickness, respectively. Additionally, tablets (n = 6) were tested for disintegration in 800 mL of demineralized water at 37 °C using disks (DT50, Saint-Louis, France). Tablet friability was determined using a friability tester described in the European Pharmacopeia (TAR 220, Erweka, Langen, Germany) at a speed of 25 rpm for 4 min. The percentage of weight loss was expressed as tablet friability.

#### 2.2.5. Binder Characterization

Wettability was evaluated with contact angle (CA) measurements using a drop shape analyzer (DSA 30, KRÜSS, Hamburg, Germany) via the sessile drop method. Pure mannitol tablets were prepared using a hydraulic press in a 13 mm diameter die (Specac pellet press, Kent, UK). In order to yield low porosity tablets (<10%), powder compression was exerted at a force of 98 kN for 1 min to avoid drop penetration bias [32]. 5 µL of each aqueous binder solution (8% *w/w*) was placed on the mannitol tablet. CAs (°) were determined as soon as the drop touched the surface of the tablet (CAMannitol_t0) and after 30 s (CAMannitol_t30). Measurements were performed in triplicate.

Binder energy plots were calculated based on force–displacement curves plotting compaction pressure against punch separation. An example of an energy plot is shown in Figure 1, where the first step includes powder particle rearrangement and packing (A–A′), as the punches move towards each other. In a second phase (A–B), the compaction pressure increases until a maximum pressure (B) with a corresponding minimal punch separation (D). The applied pressure is released in the third phase (B–C), called decompression or unloading. The areas of ABD and BCD represent the work of compression and the work of elastic recovery, respectively. The work of compaction is defined as the difference between the work of compression and the work of elastic recovery. The plasticity factor (*PF*) of a binder (%) was calculated according to Equation (3) [30,33,34]:(3)$$PF\;\left(\%\right)=\frac{Work\;of\;compaction}{Work\;of\;compression}\times 100$$

To determine the energy plots of the different binders, the powder was individually weighed (400 mg, *n* = 6), manually filled into the die, and compacted using a STYL’one Evolution compaction simulator (Medelpharm, Beynost, France). The main compaction pressure of 191 MPa was applied without a pre-compaction step at a punch speed of 13.5 mm/s. External lubrification was applied to minimize confounding of the results due to friction. A spraying time of 500 ms and an atomizing pressure of 3 bar were used as settings for the external lubrication system implemented in the compaction simulator.

Additional binder characterization (particle size distribution, dissolution kinetics, wettability, surface tension and viscosity) was previously performed and described by Vandevivere et al. [16]. The characterization data were used in the current study to examine the link between granule quality and binder attributes.

#### 2.2.6. Multivariate Data Analysis

PCA and partial least squares (PLS) were performed on various datasets. SIMCA^®^ 16 software (Sartorius Stedium Biotech, Umeå, Sweden) was used for multivariate data analysis. Prior to analysis, data were pretreated by scaling to unit variance—making the analysis independent of the units used and allowing the simultaneous analysis of quantities with different magnitudes—and by mean-centering. If needed, a logarithmical transformation was executed to non-normally distributed responses.

## 3. Results and Discussion

### 3.1. Mannitol Formulation

#### 3.1.1. Granulation Experiments

In order to study the effect of binder attributes on granule quality, it was of interest to include binders showing different intrinsic attributes. PCA (PCA_1) was executed on a dataset including 13 variables (intrinsic binder characteristics) and 11 observations (binders). These binders covered the different chemical natures of the most commonly used binders for a continuous twin-screw wet granulation process. Table 2 summarizes the binders and characterization techniques with corresponding abbreviations used for PCA. A PCA model containing two principal components (PC) explaining 51.3% and 22.6% of the variation, respectively, was fitted. The different binders’ relation according to their attributes in the loading scatters plot was illustrated by the PCA score plot. Binders clustered in the score scatter plot accordingly possessed comparable values for binder properties in the loading scatter plot. As a result, 4 main clusters were identified via hierarchical cluster analysis among the binders evaluated (Figure 2). Binders from different clusters were selected for the granulation experiments to ensure maximum variability among the binders tested: (i) SOS CO 01 and PVA 4-88, (ii) PVP K12 and maltodextrin 6, (iii) HPMC E15, and (iv) HP pea starch and PVP K90. In case binders were chemically related and located in the same cluster, only one binder was selected. In this way, the selected binders covered the distinct attribute ranges of binders.

After performing granulation experiments, the granule friability was determined and is reported in Figure 3. For each binder, a higher L/S-ratio resulted in less friable granules. This was attributed to more material dissolving in the granulation liquid, forming more solid bonds upon recrystallization of mannitol during drying [16,30]. A friability threshold of 30% was used to define the granule quality, as above this limit, granules were susceptible to breakage and attrition during downstream operations [11,16,24,26,35,36,37]. As a binder was considered more efficient when a lower amount of liquid was required to meet the 30% friability limit, the required L/S-ratio to meet this limit was used to distinguish binders on effectiveness. For PVP K12, maltodextrin 6 and HP pea starch, an L/S-ratio of 0.0675 was needed to achieve this limit, while a higher L/S-ratio (0.0800) was required for PVA 4-88, SOS CO 01 and PVP K90. For HPMC E15, an L/S-ratio of 0.1050 was needed to produce granules having a low friability (<30%). Therefore, binder effectiveness variated between the binders when granulated with mannitol: PVP K12, maltodextrin 6 and HP pea starch were defined as the most effective binders and HPMC E15 as the least effective binder. However, only a limited difference in binder effectiveness between the binder types were found. This was attributed to the high aqueous solubility of mannitol, allowing this excipient to participate in the granulation process, contributing to bond formation within the granule. Table 3 shows the average torque values measured at the L/S-ratios required to meet the friability limit for each binder. No correlation between torque and binder effectiveness was found. For PVA 4-88, remarkably low torque values were recorded for the production of low friable granules (<30%). Additionally, differences in the lowest achievable granule friability were observed between the binders at their highest processing L/S-ratios: very low granule friability of 3.0% and 2.2% was obtained with PVA 4-88 and SOS CO 01, respectively, while HPMC E15 resulted in a granule friability of 19.3%.

#### 3.1.2. Binder Characterization

Powder wetting during wet granulation is essential, as this critically influences the properties of the initial agglomerates [22]. Whether powder wetting by a binder solution is energetically favorable can be determined via the CA between solid and binder solution [38]. Accordingly, the wetting properties of mannitol were examined by measuring the CA between mannitol and the binder solutions. The CAs obtained with the different solutions immediately when the binder solution touched the mannitol tablet (t0) and after 30 s (t30) are presented in Figure 4. HPMC E15 wetted mannitol the poorest, as this binder resulted in the highest CA against mannitol at t0 and at t30. SOS CO 01, HP pea starch, maltodextrin 6 and PVP K12 resulted in the lowest CAs at t0, showing better wetting properties of mannitol compared to the other binders. However, at t30, each binder (except for HPMC E15) showed good wetting properties of mannitol (<40°), with PVP K12 resulting in the most favorable wetting, as a very low CA was obtained with this binder (<30°).

A second PCA (PCA_2) was developed using a different dataset in comparison to PCA_1. Descriptors for binder wettability on polytetrafluoroethylene (PTFE) surface were replaced by descriptors for binder wettability on mannitol tablets. Furthermore, the particle size of the binders was excluded in PCA_2 since this descriptor showed no relation with the binder effectiveness. Furthermore, only the binders selected based on PCA_1 were included. PCA_2 included 7 observations (binders) and 10 variables (binder characteristics). Table 2 summarizes the PCA variables with according abbreviations. The binder characterization data are reported in Table 4. A PCA model was fitted containing two principal components explaining 77.7% and 17.0% of the variation, respectively. The loading plot (Figure 5, right) was used to reveal the relations among the binder attributes, as it depicts if they are (positively or negatively) correlated to each other. Moreover, loadings show the importance of binder attributes towards the PCs [20,39]. The viscosity properties negatively correlated with the dissolution kinetics and positively with the wettability properties (CAMannitol_t0, CAMannitol_t30, CAbinder_t0 and CAbinder_t30). This showed that low viscous binders were linked with good binder wetting (low CA values), with good wetting of mannitol particles (low CA values), and with fast dissolution kinetics. The inverse relationship between dissolution kinetics and viscosity is widely acknowledged in literature [40,41]. The surface tension strongly impacted the second PC.

#### 3.1.3. Correlation between Binder Attributes and Binder Effectiveness

Activation of binding properties of a dry added binder requires the interaction of the binder with the granulation liquid. As mannitol exhibits a high aqueous solubility and dissolution rate, part of the granulation liquid also interacted with this excipient [42]. Accordingly, not all granulation liquid was available for binder activation.

Binders were color-coded in the function of their binder effectiveness on the score scatter plot, i.e., binders with a similar color required a similar L/S-ratio to meet the friability limit of 30% (Figure 5, left). The L/S-ratio and color of the binders are listed in Table 3. Binders requiring a similar L/S-ratio to produce low friable granules grouped on the score scatter plot (i.e., similar color), indicating that similar binder effectiveness was obtained for these binders. Accordingly, as the clustered binders had similar properties regarding the loading scatter plot, binder effectiveness was related to binder properties. Three binder clusters were observed: (i) maltodextrin 6, HP pea starch and PVP K12, (ii) PVA 4-88, PVP K90 and SOS CO 01, and (iii) HPMC E15. Furthermore, the lowest granule friability was achieved with PVA 4-88 (3.0%) and SOS CO 01 (2.2%) at their highest processing L/S-ratios, whereas the friability of granules produced with PVP K90 at the highest L/S ratio was clearly higher (9.9%). To reflect this difference in binder performance, the cluster consisting of PVA 4-88, SOS CO 01 and PVP K90 was subdivided into two sub-clusters. Different symbols were assigned to distinguish between these two sub-clusters: PVA 4-88 and SOS CO 01 (circle), and PVP K90 (triangle). The four different clusters are encircled on the score scatter plot (Figure 5, left).

The highest binder effectiveness was seen with PVP K12, maltodextrin 6 and HP pea starch, as these binders required the lowest amount of liquid to obtain the friability limit (L/S-ratio of 0.0675). These binders were clustered on the left side of the score scatter plot due to their good binder wetting, good wetting properties of mannitol, fast dissolution kinetics, high surface tension, and low viscosity. In contrast, HPMC E15, which was located on the right side of the score scatter plot, had opposite binder attributes and the lowest binder effectiveness (required L/S-ratio of 0.1050). The difference in effectiveness between these binder clusters was explained as follows: when in contact with granulation liquid, the particles of PVP K12, maltodextrin 6 and HP pea starch dissolved rapidly due to good binder wetting and fast dissolution kinetics, resulting in rapid activation of the binding properties. The fast dissolution of these binders was also positively affected by their low viscosity attributes. When activated, the dissolved binder wetted the (remaining) dry particles of mannitol efficiently, as low CAs were measured of these binder solutions against mannitol surfaces. In contrast, slower binder activation was obtained for HPMC E15, as this binder possessed slow dissolution kinetics. The binder’s high viscosity and poorer binder wetting also contributed to the slower binder activation. HPMC E15 also showed less good wetting properties of mannitol. Hence, the faster binder activation generated more interaction between the powder bed and the binder, improving the granule quality. Moreover, as a typical material residence time in the granulator is 5 to 20 s [43], more binder could dissolve in the granulation liquid when a binder had fast dissolution kinetics compared to a binder with slow dissolution kinetics. The higher amount of dissolved binder allowed to form more solid bridges during drying, yielding a lower granule friability at a similar amount of liquid.

PVA 4-88, SOS CO 01 and PVP K90 showed average binder effectiveness, as these binders required an intermediate L/S-ratio (0.08). These binders were grouped on the bottom part of the score scatter plot mainly due to their low surface tension. These binders also showed medium dissolution kinetics, low (PVA 4-88 and SOS CO 01) to medium (PVP K90) viscosity, average (PVP K90) to good (PVA 4-88 and SOS CO 01) binder wetting, and average (PVA 4-88 and PVP K90) to good (SOS CO 01) wetting of mannitol. The slower dissolution kinetics correlated with, the lower effectiveness of these binders compared to PVP K12, maltodextrin 6 and HP pea starch since binder activity during the granulation process was reduced, as less binder could dissolve in the granulation liquid. When binder activation occurred, good to average wetting of mannitol by these binders was obtained.

As illustrated in Figure 3, the lowest granule friability was achieved by PVA 4-88 and SOS CO 01 at their highest processing L/S-ratios. As these binders showed medium dissolution kinetics (PVA > SOS CO 01), a higher liquid amount positively influenced the binder hydration, probably resulting in full activation of the binder properties. When activated, the low surface tension of both binders facilitated the spreading of the droplet over the powder, resulting in a better binder distribution [44]. PVP K90 had slightly lower dissolution kinetics and a higher viscosity compared to SOS CO 01 and PVA 4-88, stressing the importance, next to the low surface tension, of fast dissolution kinetics combined with low viscosity to achieve low granule friability at higher liquid content.

For mannitol-based formulations, it was concluded that higher binder effectiveness was linked to fast activation of the binder properties. This was mainly obtained by fast binder dissolution kinetics combined with low viscosity attributes and good binder wetting. When activated, good wetting properties by the binder of mannitol also positively affected the binder effectiveness. The lowest granule friability was achieved when granulation was performed with PVA 4-88 and SOS CO 01 at the highest processing L/S-ratios, despite the binders’ intermediate attributes (in terms of viscosity, dissolution kinetics and wetting). As each binder type resulted in an efficient granulation process at low L/S-ratios, each binder type was considered suitable for continuous twin-screw wet granulation with a highly soluble formulation.

#### 3.1.4. Tablet Characterization

The tensile strength of tablets as a function of the compaction pressure produced is illustrated in Figure 6. Compaction pressures applied for tableting of the mannitol—HPMC E15 formulation showed large variability because of inconsistent die filling. Particle size distributions of the different formulations were similar and could therefore not bias the comparison of the formulations. At similar compaction pressures, tablets containing HPMC E15 resulted in low tensile strengths (<1 MPa), all other binders yielded tablets with tensile strengths higher than 2 MPa. In general, tensile strength higher than 1.7 MPa usually ensures a good tablet quality [45,46,47]. The poor tableting behavior of HPMC E15 was attributed to the binder’s lower plasticity factor (Table 5), as less plastic deformation is inherently linked to a decrease of particle bonding areas, negatively affecting compact formation with sufficient interparticulate bonds [33].

For each binder, the friability of tablets produced at the highest main compaction pressure was compliant to the European Pharmacopeia (<1.0%). However, PVA 4-88, HP pea starch, PVP K90 and PVP K12 already met the 1.0% limit at low compaction pressures. Tablets including HMPC E15 exhibited the highest tablet friability and shortest disintegration time (<3 min) over the investigated compaction pressure range, which was linked to the lower tensile strength of these tablets. Long disintegration times were recorded for PVP K90 (>10 min), which also exhibited the highest tensile strength (Figure 6) [33,34].

### 3.2. Binder Selection: DCP versus Mannitol Formulation

Whereas in the first part of the current paper, the impact of binder attributes on the granulation process of a formulation based on a highly soluble filler (mannitol) was investigated, this was previously investigated in a similar way for formulations based on a poorly soluble filler (DCP) by Vandevivere et al. [16]. Both studies indicated that binder attributes affected the binder effectiveness in a continuous wet granulation process. The selection of appropriate binders should therefore focus on binders having those attributes, resulting in the highest binder effectiveness, and on binders achieving the lowest granule friability. The critical binder attributes affecting granule formation for mannitol and DCP-based formulations will be compared below.

Efficient granulation with the mannitol and the DCP formulation required different process settings. Granule friability obtained with different binders is illustrated in Figure 7 for both formulations. For each binder, a lower liquid amount was needed to produce low friability granules when granulated with mannitol compared to DCP. Table 6 shows the L/S-ratio required to meet the 30% granule friability limit for each binder for both formulations (L/S mannitol and L/S DCP). Additionally, the L/S-range where all binders were below the friability limit was broader for granulation with DCP (0.1300–0.2300) compared to mannitol (0.0675–0.1050). As mannitol dissolves in the granulation liquid, this excipient contributes to the bond formation, hence requiring less granulation liquid to produce low friable granules. In contrast, when granulating the practically insoluble DCP formulation, bond formation within the granules solely resulted from the binder. Similar observations were made by Verstraeten et al. and by Portier et al., suggesting that a poorly soluble excipient inhibited the penetration of granulation liquid in the powder bed due to poor wetting, requiring a higher amount of granulation liquid to ensure an optimal powder-liquid interaction [11,29]. As DCP did not contribute to the granulation process, the intrinsic binder effect was more pronounced for the DCP formulation. The selection of a suitable binder is therefore considered especially important for a poorly soluble formulation [16].

The friability measured of mannitol-based granules granulated with pure water (i.e., no added binder) and with different binder solutions is compared in Figure 3. At low L/S-ratios (<0.08), it was not possible to produce granules with pure mannitol, as the barrel fill of the limitedly wetted powder bed was too high, causing machine blockage. At higher L/S-ratios (≥0.08), a strong bond formation was obtained with pure mannitol, as the granule friability was already below the 30% friability limit. The use of a higher L/S-ratio did not result in lower granule friability. In contrast, at higher L/S-ratios, the addition of specific binders to the mannitol formulation allowed to lower the absolute granule friability. However, binder selection should be performed thoroughly, as some binders resulted in a higher granule friability compared to pure mannitol (e.g., HPMC E15). In contrast to the mannitol formulation, granulation with pure DCP resulted in extremely high granule friability, regardless of the L/S-ratio. Therefore, rational binder selection was considered especially valuable for a poorly soluble formulation because only in this way, good granule quality was achieved [16]. Changing the included filler into one having a higher aqueous solubility can consequently reduce the importance of selecting the most appropriate binder.

As binder types showed different effectiveness in combination with mannitol or DCP (Figure 7), it is evident that the binder attributes which are important to guide binder selection differ in function of the solubility of the formulation. To illustrate this, a two-component PLS model was constructed, explaining 45.4% and 35.0% of the variability, respectively. Table 4 and Table 6 provide an overview of the numerical values for the X-variables and Y-variables, respectively. In Figure 8 (right), “L/S DCP” and “L/S mannitol” represent the L/S-ratio required to meet the friability limit of 30%, and “friability DCP” and “friability mannitol” were defined as the lowest achievable granule friability obtained with a specific binder, regardless of the L/S-ratio. Accordingly, the loading scatter plot was used to illustrate the relation between the binder attributes, the binder effectiveness, and the lowest achievable granule friability for both formulations. To obtain high binder effectiveness with DCP, a low surface tension (ST), good binder wetting, and good wetting of dispersive surfaces (CADCP_t0 and CAPFTE_t0) were required. Additionally, slow dissolution kinetics and high viscosity positively affected the effectiveness of binders when added to DCP. This was explained by an increase in the stickiness of the binder’s surface caused by the formation of a viscous gel layer. As observed in our previous study on DCP, the increased binder stickiness changed the powder bed’s consistency, resulting in more intensive densification and in higher cohesiveness [16]. Dhenge et al. linked a prolonged residence time with highly viscous binders, improving binder distribution [21]. Binders that resulted in the lowest achievable granule friability for the DCP formulation were located on the right side of the score scatter plot (e.g., HPMC E15) [16].

For mannitol, high binder effectiveness was correlated with good wetting properties of mannitol (CAMannitol_t0 and CAMannitol_t30), good binder wetting (CAbinder_t0, CAbinder_t30), low viscosity properties (DynamicViscosity and ViscositySlope) and fast dissolution kinetics (DissRate_t30, DissRate_t60, DissRate_t90). Binders exhibiting these properties included HP pea starch, maltodextrin 6, and PVP K12. Fast binder activation was a key attribute in granulation with mannitol, as this activated binding properties earlier in the process. This faster binder activation resulted in a longer interaction time, and subsequently, in better granule quality. Slower binder activation was linked to highly viscous binders with slow dissolution kinetics. As only a limited amount of granulation liquid was available for binder activation, it was assumed that the surface for these binders was less sticky. Accordingly, no change in consistency of the powder bed could be obtained, which also contributed to lower binder effectiveness [16].

Binders achieving the lowest granule friability can be visualized on the score scatter plot by a perpendicular projection of each binder on an imaginary line drawn through “friability mannitol” and the origin: PVA 4-88 and SOS CO 01. As the descriptors “L/S DCP” and “L/S mannitol”, and “friability DCP” and “friability mannitol” were situated on opposite sides of the origin of the loading scatter plot, it was clearly illustrated that different binder attributes affected the binder effectiveness and the lowest achievable granule friability for both formulations. It should also be noted that high binder effectiveness was not the result of one specific binder attribute but of the combination of different binder attributes, both for mannitol- and DCP-based formulations [16]. A concise overview of the binder attributes influencing binder effectiveness is addressed in Figure 9.

## 4. Conclusions

A thorough study on binder selection for continuous twin-screw granulation was performed. First, binder effectiveness was investigated for a highly soluble formulation (mannitol). Binders with different attributes were selected and were added to the granulation process via the dry addition method. As binders were extensively characterized, their attributes could be linked to binder effectiveness via PCA. Clustered binders on the PCA score scatter plot showing the same binder effectiveness. It was therefore concluded that the behavior of binders during processing depended on their intrinsic physical properties. Additionally, good tablet quality was obtained with all binders, except for HPMC yielding tablets with low tensile strength and high friability.

The second part of the current study showed that binder effectiveness depended on the solubility of the formulation caused by different requirements in terms of processability (L/S-ratio). Binder addition proved especially valuable for a poorly soluble formulation, as bond-forming solely relies on the binder. Moreover, a suitable binder needs to be selected, as the intrinsic binder effect was more pronounced for the DCP formulation. When a highly soluble excipient was used, binder addition was only required to achieve the lowest possible granule friability, as low granule friability was already obtained using the pure excipient (i.e., without binder).

Critical binder attributes influencing the binder effectiveness were identified for a poorly and highly soluble formulation: fast activation of binder properties was of key importance for a highly soluble formulation achieved by fast dissolution kinetics, low viscosity and good wetting properties of the binder. When the binder was activated, good wetting of mannitol by the binder positively affected the binder’s effectiveness. Low surface tension was demonstrated to be important to achieve the lowest granule friability. For a poorly soluble formulation, a low surface tension and a proper binder wetting combined with good wetting properties of dispersive surfaces resulted in high binder effectiveness. In addition, high viscosity attributes and slow dissolution kinetics positively affected the binder effectiveness. It is important to notice that the effectiveness of binders cannot be attributed to only one single binder characteristic but to an interplay of different properties.

The effect of the binder types with the use of a more realistic drug formulation (e.g., containing a high dosage level of paracetamol or ibuprofen) on downstream processing needs further investigation, as binder attributes impact the properties of the final dosage form. Furthermore, drug release profiles should be taken into account. This will guide a scientific-based binder selection when a formulation for processing via TSG is composed.

## Figures and Tables

**Figure 1 pharmaceutics-13-00210-f001:**
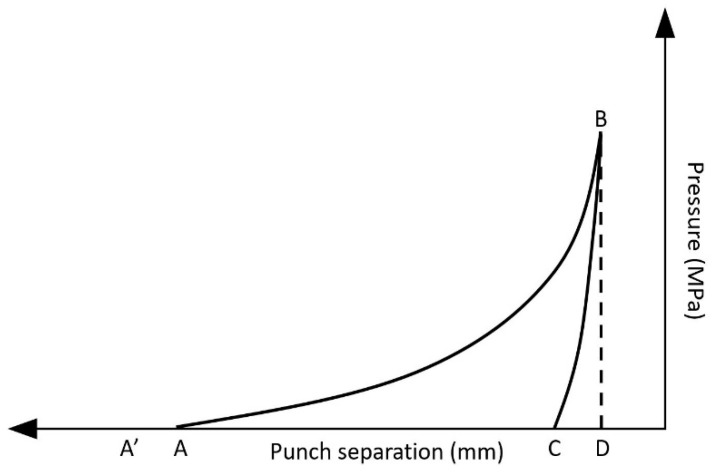
Force–displacement curve illustrating the different phases during compression.

**Figure 2 pharmaceutics-13-00210-f002:**
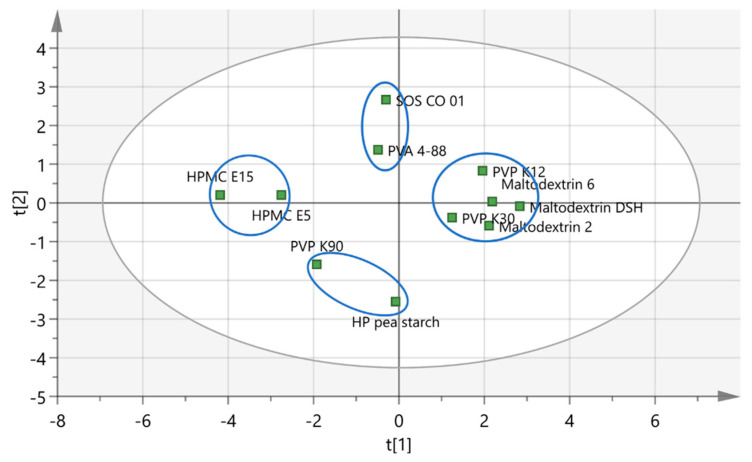
Score scatter plot illustrating principal component (PC) 1 versus PC 2 of PCA_1. Binder clusters are encircled [16].

**Figure 3 pharmaceutics-13-00210-f003:**
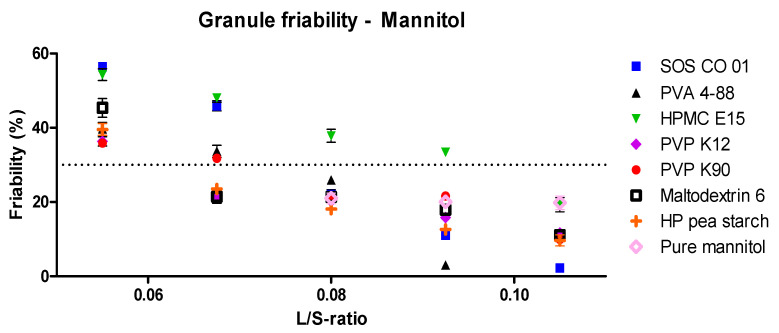
Friability of granules as a function of L/S-ratioand binder type for the mannitol formulation. The granule friability obtained with pure mannitol is also illustrated (pink diamond). The granule friability limit of 30% is illustrated with the dotted line, representing the limit to identify sufficiently strong granules.

**Figure 4 pharmaceutics-13-00210-f004:**
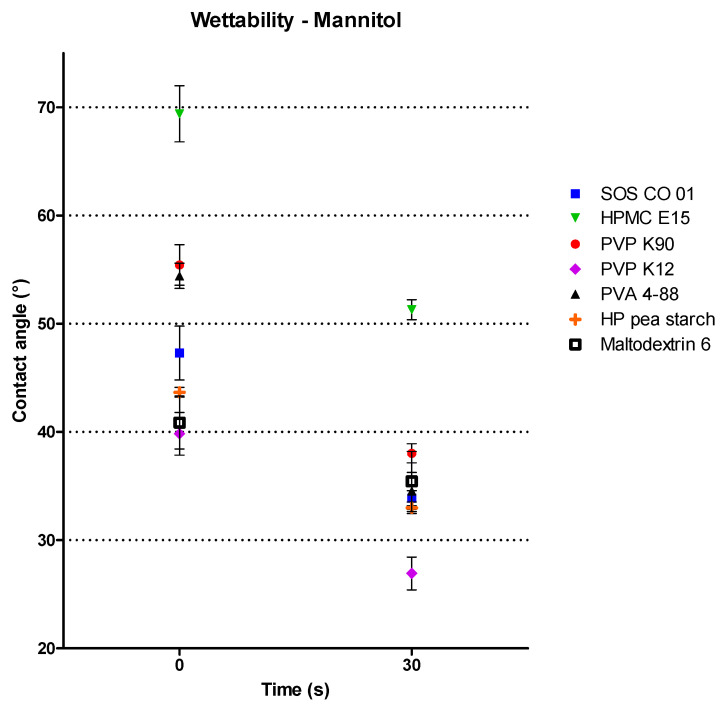
The contact angles of different binder solutions (8% *w/w*) measured against mannitol immediately after the binder solution touched the mannitol tablet (t0) and after 30 s (t30).

**Figure 5 pharmaceutics-13-00210-f005:**
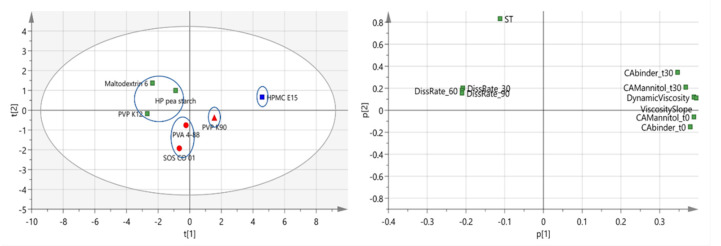
The score scatter plot (**left**), and the loading scatter plot (**right**) for PC 1 versus PC 2 of PCA_2 [16]. Binder clusters are encircled on the score scatter plot. The different colors on the score scatter plot refer to the L/S-ratio, which was required to obtain granules with a friability <30%.

**Figure 6 pharmaceutics-13-00210-f006:**
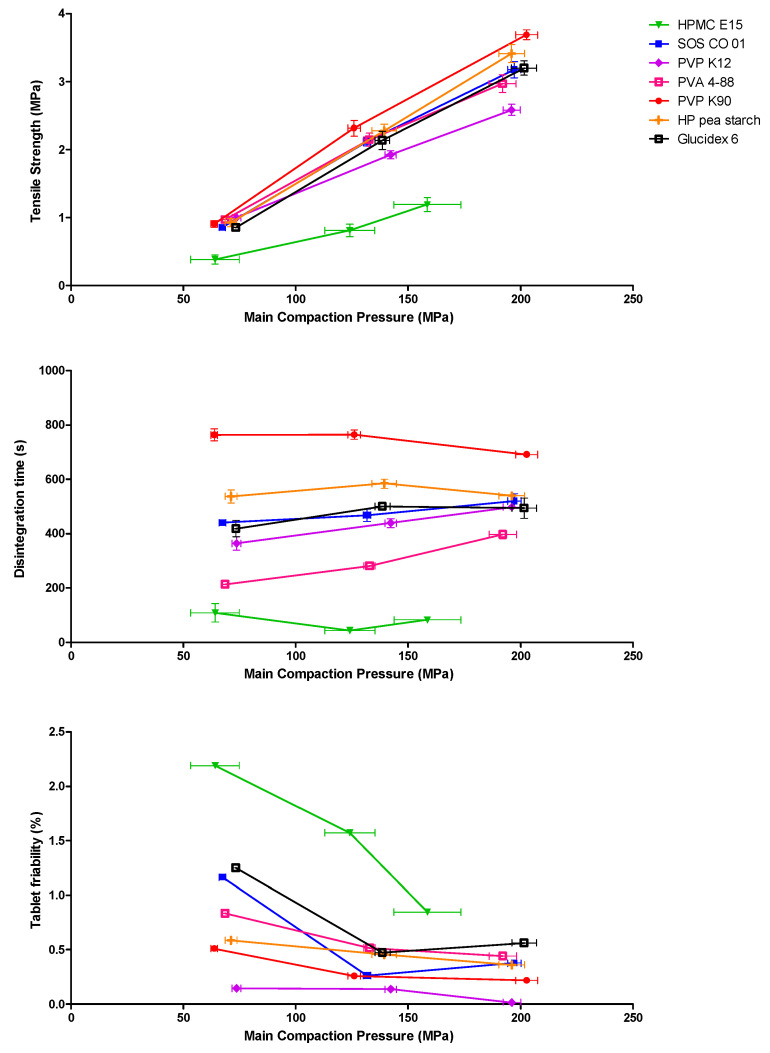
Tablet tensile strength, disintegration time and tablet friability as a function of the main compaction pressure. Tablets were obtained from milled granules produced with different binders for the mannitol formulation at an L/S-ratio of 0.08.

**Figure 7 pharmaceutics-13-00210-f007:**
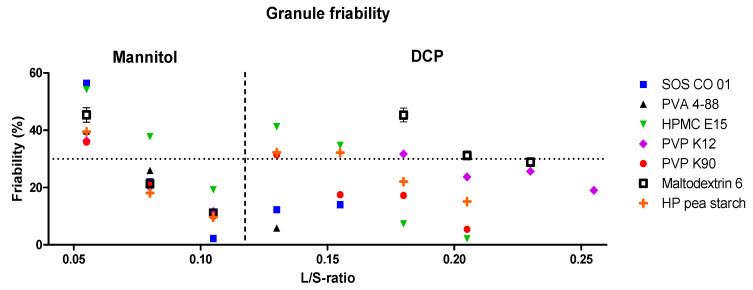
Friability of granules as a function of the L/S-ratios for the different binders illustrated for the mannitol and for the DCP formulation. The granule friability limit of 30% is illustrated with the dotted line, representing the limit to identify sufficiently strong granules [16].

**Figure 8 pharmaceutics-13-00210-f008:**
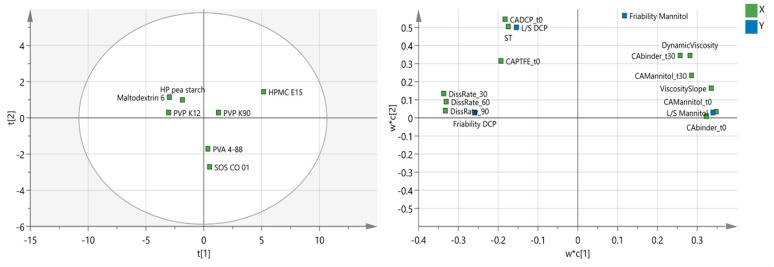
PLS model with score scatter plot (**left**), and loading scatter plot (**right**) for the first and second PC. “L/S DCP” and “L/S mannitol” refer to the L/S-ratio needed to meet the friability limit of 30%. “Friability DCP” and “friability mannitol” indicate the lowest friability that could be achieved with a binder regardless of the amount of granulation liquid [16].

**Figure 9 pharmaceutics-13-00210-f009:**
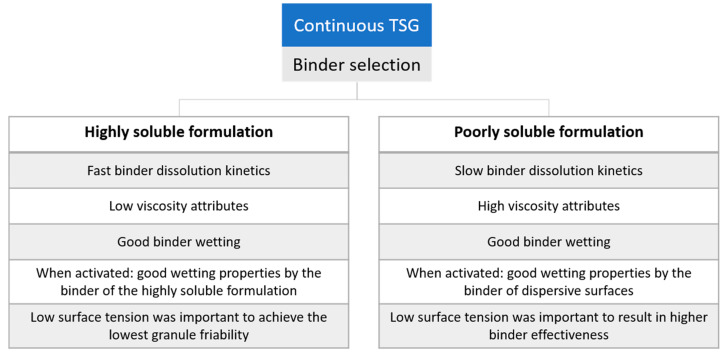
Concise overview of the critical binder attributes influencing the binder effectiveness for a poorly and highly soluble formulation.

**Table 1 pharmaceutics-13-00210-t001:** L/S-ratios used for granulation experiments of poorly-soluble filler (DCP) illustrated with different binders [16].

Binder	L/S-Ratio Used for Granulation
PVP K12	0.180, 0.205, 0.230, 0.255
PVP K90	0.130, 0.155, 0.180, 0.205
HPMC E15	0.130, 0.155, 0.180, 0.205
PVA 4-88	0.105, 0.130
Maltodextrin 6	0.180, 0.205, 0.230
SOS CO 01	0.105, 0.130, 0.155
HP pea starch	0.130, 0.155, 0.180, 0.205

**Table 2 pharmaceutics-13-00210-t002:** Summary of the binders and characterization techniques with corresponding abbreviations used for principal component analysis (PCA). Superscripts ^(1)^ and ^(2)^ indicate which binders and variables are included in PCA_1 and/or PCA_2, respectively [16].

Binder	Characterization Technique	Abbreviation Characterization Technique Used in PCA_2
HP pea starch ^(1,2)^	Particle size distribution ^(1)^	
HPMC E15 ^(1,2)^	Dissolution kinetics ^(1,2)^	DissRate_30, DissRate_60, DissRate_90
HPMC E5 ^(1)^	CA binder solution measured on PTFE ^(1)^	
Maltodextrin 2 ^(1)^	CA binder solution measured on mannitol tablet ^(2)^	CAMannitol_t0, CAMannitol_t30
Maltodextrin 6 ^(1,2)^	CA measured on tablet binder ^(2)^	CAbinder_t0, CAbinder_t30
Maltodextrin DSH ^(1)^	Dynamic viscosity ^(1,2)^	Dynamic Viscosity
PVA 4-88 ^(1,2)^	Viscosity slope ^(1,2)^	ViscositySlope
PVP K12 ^(1,2)^	Surface Tension ^(1,2)^	ST
PVP K30 ^(1)^		
PVP K90 ^(1,2)^		
SOS CO 01 ^(1,2)^		

**Table 3 pharmaceutics-13-00210-t003:** Binders with corresponding L/S-ratio needed to obtain the granule friability limit (<30%), with granule friability and torque values obtained at this L/S-ratio, and with the color assigned on the PCA score plot (PCA_2).

Binder	L/S-Ratio	Granule Friability (%)	Torque Values (Nm)	PCA Label Color (Symbol)
PVP K12	0.0675	21.7 (±1.0)	6.1 (±0.6)	Green
Maltodextrin 6	0.0675	21.4 (±1.8)	7.2 (±0.4)	Green
HP pea starch	0.0675	23.5 (±0.8)	9.5 (±0.8)	Green
PVA 4-88	0.0800	25.9 (±1.1)	0.7 (±0.2)	Red (circle)
SOS CO 01	0.0800	22.0 (±1.3)	5.4 (±0.3)	Red (circle)
PVP K90	0.0800	21.7 (±0.1)	7.4 (±0.5)	Red (triangle)
HPMC E15	0.1050	19.3 (±1.9)	4.9 (±0.3)	Blue

**Table 4 pharmaceutics-13-00210-t004:** Numerical values of binder attributes used for PCA_2 and for the PLS model. CAPTFE_t0 and CADCP_t0 were only included in the PLS model [16]. (* Values could not be presented, as tablets of PVP K12 could not be prepared due to too high ejection forces.).

Binder Characteristic (Units)	Dissolution Kinetics (%)			Wettability (°)						Surface Tension (mN/m)	Dynamic Viscosity (mPa·s)	Viscosity Slope
Abbreviation	DissRate _30	DissRate _60	DissRate _90	CAbinder _t0	CAbinder _t30	CAMannitol _t0	CAMannitol _t30	CAPTFE _t0	CADCP _t0	ST	Dynamic Viscosity	Viscosity Slope
Binder												
PVP K12	9.6	9.9	9.9	NA *	NA *	39.8 (±2.0)	26.9 (±1.5)	96.4 (±1.5)	109.6 (±3.1)	50.7 (±0.5)	1.58 (±0.01)	0.040
PVP K90	3.0	4.0	6.6	76.7 (±2.9)	56.0 (±1.0)	55.4 (±1.9)	38.0 (±0.9)	98.2 (±1.5)	84.7 (±1.3)	44.7 (±0.4)	99.48 (±1.55)	0.175
HPMC E15	0.5	2.5	2.5	79.1 (±1.7)	70.2 (±1.8)	69.4 (±2.6)	51.3 (±0.9)	82.1 (±1.1)	69.6 (±4.4)	49.0 (±0.2)	817.10 (±2.75)	0.300
PVA 4-88	5.0	6.3	7.0	67.5 (±1.3)	48.1 (±0.8)	54.4 (±1.2)	34.6 (±1.7)	82.9 (±0.4)	36.8 (±2.5)	43.2 (±0.2)	13.08 (±0.16)	0.119
Maltodextrin 6	9.7	10.0	10.0	60.0 (±3.9)	47.8 (±2.1)	40.9 (±2.5)	35.4 (±7.8)	95.4 (±1.0)	103.0 (±5.3)	66.8 (±0.2)	2.07 (±0.07)	0.053
SOS CO 01	3.8	6.1	6.1	68.7 (±2.0)	47.2 (±0.5)	47.3 (±2.5)	33.9 (±2.1)	78.0 (±0.6)	32.8 (±3.9)	30.2 (±0.2)	4.05 (±0.03)	0.085
HP pea starch	8.7	10.0	10.0	67.0 (±3.8)	57.5 (±4.6)	43.7 (±0.5)	33.0 (±0.5)	94.9 (±0.8)	95.3 (±0.5)	57.7 (±0.5)	15.67 (±0.45)	0.110

**Table 5 pharmaceutics-13-00210-t005:** Plasticity factor (%) of each binder.

Binder	Plasticity Factor (%)
PVP K12	94.72 (±0.26)
PVP K90	94.75 (±0.09)
HPMC E15	88.01 (±0.08)
PVA 4-88	94.68 (±0.04)
Maltodextrin 6	95.69 (±0.01)
SOS CO 01	96.15 (±0.04)
HP pea starch	96.38 (±0.05)

**Table 6 pharmaceutics-13-00210-t006:** Overview of the numerical Y-variables for the PLS model per binder type. “L/S mannitol” and “L/S DCP” represents the lowest L/S-ratio required to meet the friability limit (30%). “‘Friability mannitol” and “friability DCP” indicate the lowest granule friability that could be achieved with a binder, regardless of the L/S-ratio [16].

Binder	L/S Mannitol	L/S DCP	Friability Mannitol (%)	Friability DCP (%)
PVP K12	0.0675	0.2050	11.7	19.0
PVP K90	0.0800	0.1550	9.9	2.8
HPMC E15	0.1050	0.1800	19.3	2.2
PVA 4-88	0.0800	0.1300	3.0	5.9
Maltodextrin 6	0.0675	0.2300	11.1	28.9
SOS CO 01	0.0800	0.1300	2.2	14.0
HP pea starch	0.0675	0.1800	9.7	7.3

## Data Availability

Data is contained within the article.

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
