# Peer review of "Identifying Critical Binder Attributes to Facilitate Binder Selection for Efficient Formulation Development in a Continuous Twin Screw Wet Granulation Process"

_pharmaceutics, 2021, doi:10.3390/pharmaceutics13020210_

Round 1

Reviewer 1 Report

Current manuscript aims to identify critical binder attributes for binder selection of continuous twin screw wet granulation process. Considerable amount of processing and characterizations were conducted. Methods are properly documented. Assessment was based on one model API. The conclusion is weak. The authors are recommended to address the comments below to improve the manuscript.

Comments

  1. Can you summarize what are the binders studied in literature and your rationale for the selection of binders in this study? Are the binders selected representative?
  2. Page2 Line80, How do you ensure granule content uniformity? What was done to ensure that the sampling is representative? What is the sample size and sampling procedure for friability testing and others.
  3. Page3 Line102, what is the filling level in the tumbing mixer?
  4. How long does the twin screw wet granulation process reach steady state for each run with different binders?
  5. Page6 Line221, ‘recrystallisation’ of which compound?
  6. What is a typical L/S ratio in conventional process and what is a typical L/S ratio in twin screw wet granulation? Any advantage/disadvantage that was observed? By looking at the 7 binders at different L/S ratio, what do you conclude? What is the critical physical attribute? Discussion should be added.
  7. 1.4 Tablet characterization section, do you have controls – tablets made from the physical mixture of the formulation (no granulation conducted). This will tell if granulation is doing anything to your tablets and is the compaction process overpowering the granulation process –masking (to some extent) the binder’s effect.
  8. Although a lot of experimets/characterizations were conducted, I did not think that the authors made it clear in the manuscript – what are the critical binder attributes for binder selection of continuous twin screw wet granulation process. The conclusion is very weak in the manuscript, quoted here: “In conclusion, optimal binder selection in continuous wet granulation should rely on the critical binder attributes influencing the granulation process.”

Reviewer 2 Report

The manuscript was very conceptually written. All figures and tables were easily understandable and the language was good. I have no changes in manuscript.

Author Response

The authors want to thank the Reviewer for the comments.

Reviewer 3 Report

In this work, the authors identified that binder attributes affecting the wet granulation process of a highly soluble model excipient (mannitol). However, the quality of the manuscript is deficient in terms of pharmaceutical sciences. Researchers have to describe the results and discussions based on quantitative numbers and evidence, but they have qualitatively described the results in many cases. Additionally, there was no uniformity for terms, and the definition of terms was unclear in terms of pharmaceutical sciences. Specifically, L/S ratios are not suitable concept for preparing the pharmaceutical granulation form. Authors should use the weight ratio for setting the composition of pharmaceutical forms. The release and disintegration behavior of a model drug with paracetamol or ibuprofen, which are very common and having efficacy, should be considered to express the conceptualization of this paper. In fact, the concept of friability is limited for tablet formulations, and its standards and test procedures  are established by USFDA and USP. I cannot clearly understand the concept for the PLS and Multivariate data analysis. These kinds of data are not the main thing but the supplementary thing for describing the performance of pharmaceutical formulation. I would concern that experimental data produced by intentionally using concepts ‘granule friability’ could make readers to confuse the definition for pharmaceutical friability. After carefully reading the manuscript, therefore, I have concluded this paper should be rejected and encouraged the resubmission of the manuscript in other journals.

Reviewer 4 Report

Dear Authors,

Scientific paper is well written. It provides useful tool for binder selection in process of twin screw wet granulation process.

Best regards

Author Response

(The authors gave the same response as above.)

Round 2

Reviewer 1 Report

The authors have thoroughly addressed comments. The manuscript is significantly improved. 

Reviewer 3 Report

In this work, all the concepts and author's responses are based on the other peer-review journals and papers. However, I cannot even support the acceptance of this paper. This paper makes readers confuse to understand the formulation studies and the investigation of critical quality attributes for the selection of excipients. The author should set any model drug and the factor to compare the binder effectiveness. Descriptions related to the development method or PCA data interpretation are not clear, and the expressions for subjects or words as a whole have serious deficiencies from the pharmaceutical science perspective. After carefully reading the revised manuscript, therefore, I have concluded this paper should be rejected and encouraged the resubmission of the manuscript in other journals.